# Mapping Academic Literature on Governing Inclusive Green Growth in Africa: Geographical Biases and Topical Gaps

**Adam Cooper** [1,*], **Chipo Mukonza** [2], **Eleanor Fisher** [3] , **Yacob Mulugetta** [1], **Mulu Gebreeyesus** [4], **Magnus Onuoha** [5], **Abu-Bakar Massaquoi** [6], **Kennedy Chigozie Ahanotu** [7] and **Chukwumerije Okereke** [8,9]

1   Department of Science, Technology, Engineering and Public Policy, University College London, London WC1E 6BT, UK; yacob.mulugetta@ucl.ac.uk
2   Department of Economic and Management Sciences, Tshwane University of Technology, Pretoria West, Pretoria 0183, South Africa; chiponyam@gmail.com
3   International Development Department, University of Reading, Reading RG6 6AH, UK; e.fisher@reading.ac.uk
4   Ethiopia Development Research Institute, Addis Ababa 2479, Ethiopia; mulu.yesus@gmail.com
5   Sustainable Energy Practitioners Association of Nigeria, Abuja, Nigeria; Magnus1@gmail.com
6   Institute of Environmental Management and Quality Control, Njala University, Njala, Moyamba District, Sierra Leone; massaquoiabubakar@gmail.com
7   Society for Planet and Prosperity, Abuja, Nigeria; kennygold500@gmail.com
8   Department of Geography and Environmental Science, University of Reading, Reading RG6 6AH, UK; c.okereke@reading.ac.uk
9   Centre for Climate Change and Development, Alex Ekwueme Federal University Nigeria, Abakaliki, Nigeria
*   Correspondence: adam.cooper@ucl.ac.uk

**Abstract:** A strong indigenous capacity for credible, salient and legitimate knowledge production is crucial to support African countries in developing their economies and societies inclusively and sustainably. In this article, we aim to quantify the current and historic capacity for African knowledge production to support the green economy in Africa, and identify important topical gaps. With a focus on topics relating to Governing Inclusive Green Growth in Africa (GIGGA), our research mapped how much Africa-focused research is being produced, from where and which African countries have higher or lower supply; and the topical focus of the research, mapping it against the African GIGGA policy discourses visible in government strategies. To do this we undertook a systematic review using a two-stage process, mapping the literature for GIGGA. This resulted in 960 verified citations. Content analysis of core metadata and article abstracts enabled mapping of the research focus. The analysis revealed a significant role for South Africa as both the pre-eminent producer of GIGGA literature as well as the geographic focus of GIGGA research, with Nigeria, Ethiopia and Kenya representing emerging loci of credible, African-relevant knowledge production. Topically, there was a strong emphasis on development, policy and environment while topics important for growth that is inclusive in character were infrequent or absent. Overall the results reinforced the view that investment is needed in research on inclusive green growth, linked to capacity building for knowledge production systems in Africa. Furthermore, from a policy perspective, policy makers and academics need to actively explore best to collaborate to ensure that academic research informs government policy.

**Keywords:** green growth; green governance; inclusive; Africa; capacity building

## 1. Introduction

Africa is at a crucial point in its developmental history. After decades of economic decline, it experienced the fastest economic growth rate of any continent in the early 2000s before appearing to slow down again in recent decades [1]. Against this background, while acknowledging the serious challenges of planning, coordination, capacity, finance and technology, commentators have noted that Africa is uniquely placed to exploit opportunities from the green economy to achieve inclusive sustainable economic transformation [2–5]. In this line, several African countries, including Rwanda, Ethiopia, Kenya and South Africa, are taking action under ambitious national plans to decouple economic growth from environmental pressures and 'leapfrog' to green sustainable economies thus avoiding inefficient technologies and products in preference for clean and resource efficient ones, thereby avoiding unsustainable lock-in [6]. Such fostering of economic growth and development while ensuring that natural assets continue to provide the resources and environmental services on which well-being relies, is commonly referred to as 'green growth' [7–11] and here we use the phrase Governing Inclusive Green Growth in Africa (GIGGA) to incorporate attention to issues of governance and inclusivity [12,13].

In recent years, green growth and the green economy have emerged as one of several paradigms for fostering a growth path that integrates and reconciles economic, social and environmental objectives [14]. Indeed, green economy strategies in several African countries are intended to respond to the pressing challenges of social equity, resource efficiency and ecological concerns in a coherent and integrated fashion [11]. This requires development policies that are responsive to resource efficiency and employment creation requirements [15]. Such policies imply questioning gross domestic product (GDP) growth as the sole measure of progress and rethinking how growth interacts with wellbeing [16]. The original impetus for promoting this agenda can arguably be traced back to the 1987 Brundtland Report "Our Common Future" [17,18] but has been given more recent urgency by four key international agreements, which were negotiated in parallel and all adopted in 2015: The Agenda 2030 for Sustainable Development, the Paris Agreement on Climate Change, Financing for Development, and African Union Agenda 2063. Each offers a vision for inclusive, sustainable, resilient and low carbon development. They alerted policy makers to the reality that future policy and planning will need to take place in the shadows of shrinking resources and changing climate, and so radical paradigm shift will need to take place to ensure higher wellbeing without depleting natural resources [19]. Indeed, it has been argued that several of the planetary boundaries that keep the planet hospitable to modern life, have now been crossed [20].

A key message emerging from these international agreements, most prominently Agenda 2030, is that environmental burdens and social costs are disproportionately shouldered by the poorest and powerless. Hence, the central mission of sustainable development is not about shifting burdens to parts of the world where the cost of production may be 'low' or 'legally permissible' but rather reducing these burdens overall. For a continent such as Africa with high poverty levels and significant inequalities the need for burden reduction and for inclusive growth is pressing. This is coupled with recognition that the world has seen major improvements and cost reductions of green(er) technologies, which has made it possible to experiment with new business models to deliver services since reducing the need for costly transmission stimulates the aspiration for needs-based services that can be delivered at speed.

Despite ambitious seeds of change, research on inclusive sustainable economic development in Africa is limited and hence the dynamics of this potential paradigm-shifting phenomenon are poorly understood. We argue here that for Africa to benefit from green growth, not only must that green growth be inclusive it must be underpinned by an effective knowledge system. Bearing this in mind, Cash et al. [21] make the case for the importance of mobilising knowledge systems for sustainable development - namely, the interaction between research and governance institutions that underpins the effective creation of knowledge for policy making. Three characteristics of scientific information production are identified that increase its effectiveness "in influencing the evolution of social responses to public issues" [21]. These comprise salience, the relevance of the scientific output to those in

governance positions; credibility, the quality of the scientific evidence generated; and, legitimacy which refers to the perception of the research process generating the evidence and its need to be seen as respectful and fair to different actors involved in the research.

Implicitly, Cash et al.'s [21] notion of legitimacy might indicate that research conducted by researchers based on the continent is better placed to be respectful and fair to the various actors and stakeholders, since awareness of local socio-economic conditions is seen as an important pre-cursor to undertaking respectful and fair research. This line of argument is echoed in sustainability research that identifies the need for effective practice in knowledge co-production [22,23]. The quality of relationships, including need for positive and trust-based relations, is also found to be important for connecting science to governance in environmental research [24]. In this respect too, locally based researchers who hold a critical understanding of context may be best placed to build these relations [25] contributing to legitimacy.

This leads us to present a systematic search conducted to gain a better understanding of knowledge capacity with respect to GIGGA across the Continent. The search aimed to answer questions brigaded around the three features of credibility, legitimacy and salience: Credibility (1) How much research on GIGGA is being produced and which countries have higher or lower supply of GIGGA research? Legitimacy (2) Where is the research being produced and how much collaboration with Africa-based researchers is there? Salience (3) What topical focus can be observed in the research and how well does that fit with African policy discourses on GIGGA?

## 2. Knowledge Systems for Governance of Inclusive Green Growth in Africa

In relation to Africa, important questions arise regarding how well set up the knowledge systems are to deliver credible, salient and legitimate research to local policy makers. Focusing on climate change research, a previous study has documented the relative lack of research capacity in African research institutes via bibliometric analysis [26]. Their evidence suggests that 56.5% of publications had a first author based in the country the study was about. For the top 20 most productive countries (based on institutional affiliation of the first authors) for climate change research only includes one African country (South Africa, 16th) and no major African countries when population size is factored in. Similarly, Kiparsky et al. [27] found that climate change research tended to focus heavily on North America, Global and European geographies, with Africa and Asia following. In their analysis of the Intergovernmental Panel on Climate Change (IPCC) reports Ho-Lem et al. [28] suggest that only 4% of the authors were African and when population was considered were the second most underrepresent geography by continent after Asia. Nevertheless, as Confraria and Godinho [29] point out, African research output has—since 2004—been increasing faster than the total output for the world which, as argued above, provides the potential for a foundation of local knowledge that can effectively inform policy making.

While there are apparently no published studies exploring academic capacity for green growth research in Africa, two studies of the academic literature with a focus on ecosystem-monitoring shed more direct light on both patterns of research to support green growth generally and African research in particular. Wang et al. [30] undertook a bibliometric analysis exploring approaches to 'low carbon development transformation'—a phrase synonymous with green growth. They list the top 20 productive countries in a similar manner to Pasgaard above, but on the topic of low carbon development, no African country is visible. This indicates that African research on green growth is clearly at a low level internationally but given research growth in recent decades merits a specific analysis to reveal emergent patterns. Their study focuses on mapping existing research and identifying future topical directions. They identified clusters around climate change and renewable energy as central to low carbon research. They see the topic of "smart grids" as one focus for future research. The topical outcomes here can be compared with the related analysis we describe for Africa below. We return to this in the Discussion.

Yevide et al. [31] undertook a bibliometric analysis of research in Africa related to ecosystem monitoring. Ecosystem monitoring can be seen as an important activity with regard to understanding the implications of development or growth on natural systems and so is a useful analysis related to green growth research in Africa. They analysed over 1400 publications covering 1987–2014, showing significant growth in output around 2000, consistent with the findings from Confraria and Godinho [29]. They also map countries of the author's institutional affiliation, showing not only the typical skew towards authors being based in the US/Europe, but African authorship heaving skewed to South Africa, followed by Southern and Eastern Africa. However, the main focus here is around research to inform policy intervention that promote green growth and so while useful, the study serves to highlight the gap in the literature we aim to address here.

In order to understand how well-placed Africa research is to support and influence decision makers, we need to understand how legitimate and salient the most credible research on this topic is, as defined by Cash et al. [21] as based on research quality. Since the most widely regarded proxy for research quality is publishing in a peer-reviewed journal, we aim to map peer-reviewed research. However, in line with Confraria and Godinho [29], we recognise that other means of defining quality (and therefore credibility) exists. For example, in policy spheres perceptions of quality may be interconnected with the international credibility of the organisations producing the data. Likewise, we acknowledge debate that publications are not a perfect measure of scientific production and related issues of research quality. However, since we are assessing the capacity to influence policy and governance, academic knowledge is likely best placed as a proxy for research quality. Understanding the pattern of research with a specific country focus will reveal where the stronger resources bases are for credible supply of salient research across the continent. Put another way, the argument of our paper is that the more–and better–data we have about green growth, the easier it is to identify research gaps and make informed decisions about the future of green growth governance. In addition, mapping out literature helps researchers to understand the limitations of our tools [32].

For salience and legitimacy, we aim to map the topical coverage across the concepts of GIGGA to see what kinds of terms are most commonly used to describe research in this area. Their salience is then judged in relation to key national policy documents in this area for three case countries – Ethiopia, Kenya and Nigeria. Legitimacy is estimated based on the authors' institutional location. To reiterate the logic set out above, we argue that being located in the same country is provides a stronger basis for effective co-production of scientific knowledge, itself a widely regarded mechanism for research-policy influence.

Research on geographic global biases with respect to African academic research is not entirely new [26,29,31,33]. This study aims to contribute to this literature by identifying where research in this area is produced, and the continental patterns of focus for relevant studies. In addition, we will look at the aftermath of the seminal (for thinking on sustainable development) 'Our Common Future' also known at the Brundtland Report [17], conferring a longer lens than previous research [18] and identify topical gaps in the GIGGA framework.

## 3. Methods

The approach applied here is bibliometric in nature and akin to a systematic review (without statistical meta-analysis) or a rapid evidence assessment (as used in the UK [34]). The key differences are that the approach was not to look at publication patterns within journals or by author or author-networks but to look at what countries on the Continent were producing research on Africa, what countries were the focus of this research and what topics featured most in the literature. For feasibility with regard to the scope of time and resources, we also took a relatively narrow approach to finding relevant literature by only using a single database as the source, rather than using multiple databases and following up references in key articles or books or gathering recommendations from experts across the fields.

A visual representation of the overall approach is provided at Figure 1. This shows the general stages from keyword definition, search strategy and mapping approach. Further detail on each of these steps is provided in the following text.

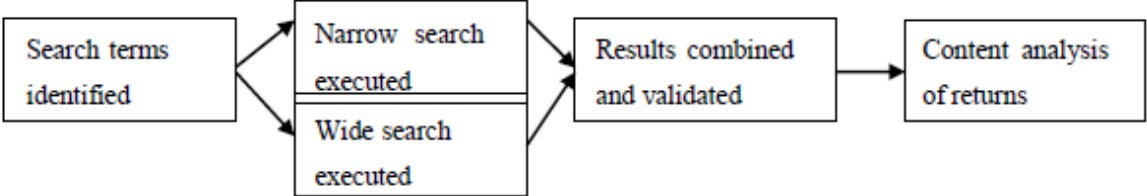

**Figure 1.** Scheme showing the major stages of the literature mapping and content analysis.

*3.1. Search Strategy*

A process of co-production governed our search strategy and framed the decisions we made over approach. The project was collaborative between academics and policy researchers across Africa and the UK (22 in total); a sub-group of eight people conducted the review reported here, with feedback from the wider group. This intercontinental academic–policy coproduction was viewed as a microcosm of the type of activity that is required to help lift research output more rapidly in a direction in which it can inform green growth policy. The nature of the collaborative, intercontinental approach for the project meant that a highly efficient search strategy was needed to maximize the time of collaborators.

As discussed above, we decided early that mapping the academic literature on GIGGA was the primary goal because it is considered that academic literature represented credible research in this area, rather than mapping wider policy-relevant studies of varying quality from outside the academic domain. Taking this assumption we agreed an approach to map studies indexed in abstracting databases to make mapping of the literature more efficient but also to utilize the filtering mechanisms those databases use to choose which journals to include. In a sense, major online academic databases like Scopus and Web of Science reflect the mainstream model of academic knowledge and so what data they comprise is a good representation of what the mainstream academia focus on. For efficiency, we chose Scopus as the database to use as this has a wider coverage compared to Web of Science's better historical records but also a more extensive exporting facility regarding metadata.

In order to map the literature in a way that gave fair representation to the range of research generated in relation to GIGGA, and to do so with maximum efficiency we decided on a two-stage approach. The first stage was a narrow search of Scopus focused on terms synonymous with 'green growth'. The logic here was that returns with this search would be directly relevant to the core topic areas of the project since they use the same terminology, and the chances of false positives would be very low. At the same time, we recognized that there are likely many research studies undertaking work in the area of GIGGA that do not use the 'green growth' or similar framing but were important to capture. This meant we undertook a wider search that used broader synonyms for green and growth. To ensure the search was not too broad and unmanageable we also ensured that any research returned also had to have some mention of topics aligned with governing and inclusiveness. The 'narrow' then 'wide' search strategy demanded the creation of two sets of search terms for each. We explain what these were and how they were decided below. Searches were limited to title, abstract and keywords in the Scopus records. Only 'article' document types were included, as they represent the main peer-reviewed way of publishing studies in any domain.

3.1.1. Scope and Exclusions

The 1987 Brundtland Report [17] was used as the cut-off date for publications to include as this was a watershed publication widely seen as providing the modern-day point at which the green economy agenda in Africa was identified [35,36]. This gives rise to a focus on core concepts captured by the concepts of governance, inclusion and green growth understood within a disciplinary academic

perspective of social and economic sciences, rather than medical or physical sciences. Literature therefore classified as related to health or physics and astronomy were excluded. In part this is due to the clear division between the focus of green economy and health, but also to reduce the number of false positives, especially in the wider search, and therefore reducing the validation burden on the research team. In Scopus, this was operationalized as excluding any research categorized under the following subject areas:

- Medicine
- Immunology and Microbiology
- Biochemistry, Genetics and Molecular Biology
- Nursing
- Health Professions
- Pharmacology, Toxicology and Pharmaceutics
- Dentistry
- Physics and Astronomy

After initial runs of the search we discovered that significant numbers of false positive, irrelevant papers about 'greenstone' mineralization were being returned, so to reduce false hits, the search term was adjusted by adding 'NOT greenstone*' specifically to exclude this word and reduce the false positives.

### 3.1.2. GIGGA Term Identification

*Defining Africa:* The simplest part of identifying relevant search terms was the dictionary of terms for defining Africa. Only African country-level name mentions plus variations on 'Africa' were explicitly searched for. This resulted in 60 separate search terms for the Africa element of GIGGA – 59 country names, including both English and French version of Ivory Coast and all minor states such as Mayotte, Saint Helena and Sao Tome and Principe. For the full list of search terms defining African countries, see Annex A.

*Narrow search terms:* Given the goal of mapping all the academic research relevant to GIGGA, the obvious approach to finding such literature is to focus on the actual terms within the GIGGA phrase. Since we were interested to understand how much had been written (or not) on governance and inclusion in this broad domain, and what other topics were emerging, the focus rested on the 'green growth' aspect of GIGGA. A pilot search of Scopus using just "green* growth" together with the Africa search terms and exclusion criteria resulted in only 19 hits. This was deemed too narrow for our purposes, especially given the common use of other related terms in this field such as 'green economy'. This gave rise to a set of terms directly linked to the 'green growth' agenda, which with wildcards gave rise to the following five search terms.

- Green* growth
- Green* econom*
- Green* development
- Green* productivity
- Resource efficien*

A search of Scopus was then conducted with these five terms as set out below, coupled with the Africa search terms and exclusion terms set out above. This resulted in 90 references being identified. Recognising the limited number of returns might easily be a consequence of the particular 'Green' framing, the team agreed to undertake a wider search that captured 'non-green' framings for these topics. The search term used can be represented as:

In title-abstract-keyword ["Green* growth" or "Green econom*" or "Green development" or "Green productivity" or "resource efficien*"] and [Africa or … ] and year published [after 1986] and document type [Article] and not subject area [Health-related fields or Physics and Astronomy]

### 3.1.3. Wider Search Terms

The strategy for the wider search combined the need to include wider terms that are likely used in reference to green economy concepts while simultaneously preventing the search from becoming so wide the number of returns would be unmanageable by the team. To balance this, the wider search was targeted at finding studies that spoke directly in some way to every element of the Governing Inclusive Green Growth (GIGG) field. This meant defining small dictionary of search terms for each of the other four elements of GIGG alongside the Africa terms described earlier. The team agreed on the following set of 55 terms to capture the range of meanings under each element of GIGG as set out in Table 1.

**Table 1.** Showing the 55 GIGG search terms generated by the research team, grouped by each element of GIGG.

| Governing | Inclusive | Green | Growth |
|---|---|---|---|
| Governmen | Inclusi* | Environment* | Natural capital |
| TGovernance | Participat* | Sustainability | Green capital |
| Democra* | Owner | Ecologic* | Develop* |
| Policy | Social responsibility | Ecosystem service* | Food security |
| Politic* | Social enterprise | *Forest* | Water security |
| Minist* | Co-operative | Agriculture | Energy security |
| Local authority | Well-being | farm* | Transformati* |
| Public sector | Quality of life | Climat* | GDP |
| Private sector | Living standards | Emission* | GNP |
| | Equity | carbon | Welfare |
| | Gender | Renewable* | Externalit* |
| | Women | REDD* | |
| | Low Income | INDC* | |
| | Rural | Recycl* | |
| | Urban | Pollution | |
| | Grabbing | Fossil fuelErosion | |
| | Exclusi* | | |
| | Poverty | | |

Note: phrases were always searched with quotation marks, e.g., "local authority".

While the terms beneath each element of GIGG are by no means exhaustive, the goal was to include terms that were central to core aspects of the different elements of GIGG. These are expanded on below.

*Governing*: Here the focus is on public governance via standard formal democratic or related political processes rather than the governance of private enterprise. In addition, the level of analysis is at structures ('government', 'local authority', 'public sector', 'private sector', 'minist*') and processes ('governance', 'policy', politic') rather than individuals or their activities. The term 'govern*' was avoided due to the issue of it generating a lot of false positives due to its common use in science with respect to processes 'governing' outcomes. The more specific variants were used instead.

*Inclusive*: Here the terms cover three different aspects of inclusion: the core meaning of the term around enabling engagement and decision-making capture both by core terms 'inclus*' and 'participat*' but also the means by which inclusion can be promoted ('social responsibility', social enterprise', 'co-operative') the benefits related to inclusion ('well-being', 'quality of life', 'living standards', 'equity') the target populations for whom inclusion is important ('gender', 'women', 'low income', 'rural', 'urban') and negative activities or outcomes related to prevention of inclusion ('grabbing', 'exclusi*', 'poverty').

*Green:* Core meanings of 'green' were captured with common terms ('environment*', 'sustainability', 'climat*', 'carbon', 'renewable*', 'recycl*', 'ecologic*') as well as both key sectoral (*forest*, agriculture, farm*) and governance terms (REDD*, INDC*, ecosystem service*). Negative concepts related to undermining progress towards achieving green goals were also included ('pollution', 'fossil fuel', 'erosion').

*Growth*: A range of terms related to the underpinning ideas around growth were included, especially where they related to environmental sustainability or were related to economic concepts capturing core resources ('natural capital', 'green capital', 'water security', 'food security', 'energy security') as well as more general synonyms for growth ('develop*', 'GNP', 'GDP'). Benefits and impacts related to growth and economic activity were also included ('welfare', 'externalit*').

### 3.1.4. Search Term

The overall approach for the search term can be characterized as:

In title-abstract-keyword [**Governing** OR . . . ] and [**Inclusive** or . . . ] and [**Green** or . . . ] and [**Growth** or . . . ] and [**Africa** or . . . ] after 1986, and not subject area [Health-related fields or Physics and Astronomy] and document type [Article]

This meant that citations returned had to have, in title, abstract or keywords, one mention of any term within each of the five GIGGA areas. Note that terms in bold indicate search term *category* not actual search terms.

### 3.1.5. Data Handling and Verification

Both the narrow and wider searches were carried out at the same time in April 2018. After running the search term, all citations were downloaded into a CSV file, then loaded into an MS Excel file for data handling. Where necessary, the citations were reviewed for validation purposes. For the narrow search, validation was carried out by one member of the team, due to the overall low number of returns. For the wider search, validation was split across the 7 members of the international team evenly, and then inclusion rates compared to ensure reasonably consistency across scorer. Final validation was done in excel using the text search function to ensure at least one instance of an African search term was present in the title or abstract, and at least one mention of the GIGG search terms was evident. After the two searches were verified, the lists of citations were combined and assessed automatically for duplicates using the conditional formatting function in MS Excel to highlight cells containing the same article titles. Duplicate found through this process were removed.

### 3.1.6. Mapping the Content and Meta Data

Once a single, final validated list of citations had been generated, the electronic identification numbers attributed to each record by Scopus were used to rerun the search in Scopus. The resulting search enabled the download of the refine values Scopus produces including counts of institutional and country affiliation. These data were used to map the author institutional geography in a heat map via the free online tool openheatmap.com.

Further topical mapping of the content of the studies was done by using the MS Excel automated text search function to find single instances of particular terms. We did this for all the search terms used originally, enabling us to map the country focus of the studies reported, and the degree to which different search terms within categories of GIGGA appeared in the same citation.

These different processes enabled us to map the geographic biases of the GIGGA research both in terms of author institutional country affiliation, and in terms of the country focus of the research. By mapping the instances of GIGGA terms we could also see what aspects of GIGGA are more commonly focused on and where are there gaps.

## 4. Results

### 4.1. Number of Returns and Validation

Overall the searches resulted in 2574 hits which, following validation were distilled down to 960 unique and verified GIGGA citations. The different breakdowns by search approach are set out below.

### 4.1.1. Narrow Search

The narrow search returned 85 hits. Review of these 85 citation's titles and abstracts revealed 12 papers that were outside the scope of GIGGA. Most of these false positives were related either to plant studies where the term 'resource efficien*' triggered irrelevant hits or 'greenfield development' studies. Removing these 12 studies resulted in 73 verified GIGGA studies from the narrow search, an 85% hit rate.

### 4.1.2. Wider Search

The wider search returned 2489 hits. The citations were ordered by title name and then divided into eight equal groups of 311 citations and then sent out to the team members. Each team member was instructed to mark the citation as 'include' only if the title and abstract referred to elements of green growth in Africa—that is explicitly forms of sustainable development, awareness of impacts on ecosystems, degradation of resources etc., and improved the livelihoods of citizens in African countries. Papers on agricultural development that did not refer to issues of sustainability were therefore excluded, and this formed the major bulk of exclusions. Anything that referred only to international conferences in an African country but were not about that African country were excluded. This means that research in an African country that referred only to governance and inclusion were excluded.

Coding resulted in 887 verified citations. Reviewing the inclusion rate revealed an average of 37% overall average across reviewers, with a range of 9–68%. Six of the eight bundles had inclusion rate between 23 and 42% so the two outliers were reviewed by Cooper by sampling citations. These reviews showed no obvious difference in the application of the criteria suggesting the outlying inclusion rates are likely down to variation based on the alphabetic ordering of citations by titles creating clusters of related topics.

### 4.2. Countries of Author's Affiliated Institutions

A key aim of the mapping was to establish the geographic locus of knowledge creators about GIGGA. As elaborated above, the underlying hypothesis presupposed that Africa-based authors would be in a better position to research African green growth issues than those outside Africa. In the first instance, we examined the directly available data from Scopus' refine value export file which counts per paper the countries of the author's institutional affiliation. So here, a count of 10 for South Africa would indicate 10 citations that have at least one South Africa-based author. Since citations often have more than one author citations can count multiple times where there are authors based in different countries.

The database revealed a total count of 1316 country-citation associations. Overall 48% of associations were with African countries, the remainder non-African. South Africa-based authors were represented on 24% of the citation-associations or 32% of all citations in the GIGGA database. The next two most common countries were the UK (12% of citation associations, 16% of all citations) and the US (9% and 13% respectively). Over 60% of all the citations in the databased carried authors based in one of these three countries. Figure 2 shows the geographic spread of citation associations globally. Within Africa, Tanzania and East Africa more broadly are emerging centers of GIGGA expertise outside of South Africa, with Nigeria also a significant actor.

Deeper analysis of the authors' institutions' geographic location was undertaken to understand what proportion of citations were by authors who were i) all based inside African countries, ii) all based outside African countries, and iii) mixed inside and outside Africa. Country affiliation data for 80 citations did not exist, so this analysis was carried out on the remaining 880 citations. We found that 349 citations (40%) had authors who were just based in Africa, 315 (36%) had authors who were just based outside of Africa, and 216 (25%) had mixed authorships. This means that African-based authors were involved in the majority of citations (75%), which clearly implies a healthy basis for improving the knowledge production relevant to the continent. However, if South Africa is removed from the

analysis, the picture changes dramatically. The total number of relevant citations drops to 663. Of those, only 132 citations (20%) are from Africa-only based authors and 145 (22%) mixed Africa-based and outside-Africa-based. This means that the majority of GIGGA research excluding those with South Africa-based authors is by authors based entirely outside of Africa (386 citations, 58%).

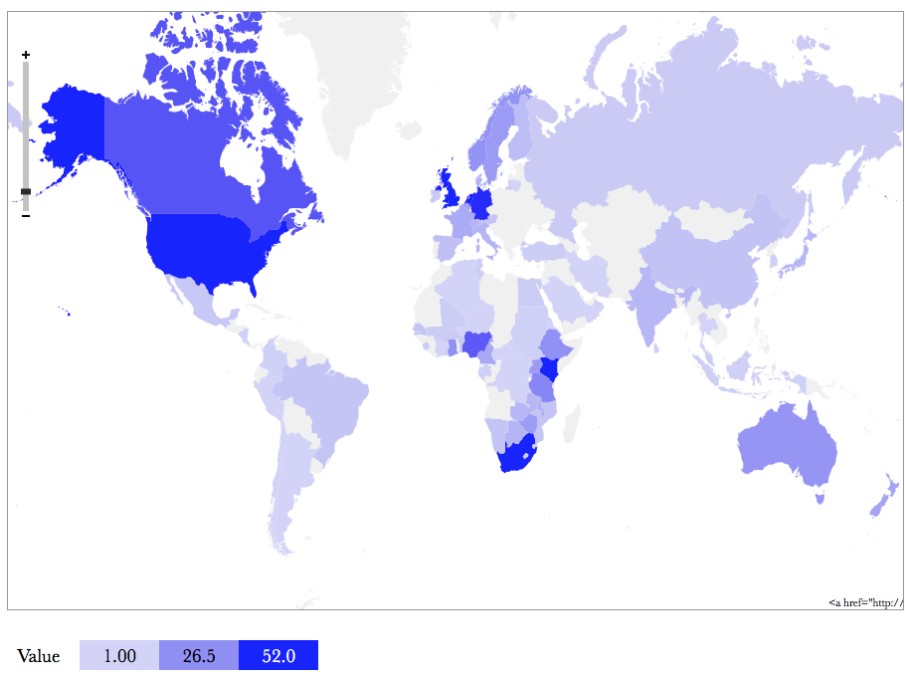

| Value | 1.00 | 26.5 | 52.0 |

**Figure 2.** Schematic heat map of authors' institutions' country location shown in a standard Mercator projection to ease interpretation and country identification. The darker the blue, the greater number of authors registered as part of an institution in that country.

### 4.3. Country Focus of GIGGA Research

The institutional geography is important in understanding what kinds of perspectives are being applied to African countries GIGGA research, but equally important are the actual geographic focus of research. Without research focused on specific countries it is difficult for countries to benefit from the application of expertise. As with the institutional geography analysis the counts here reflect the number of citations in the title or abstract of which a country is mentioned.

Since citations can deal with research across multiple countries the total can be above the number of papers overall. In total 1217 references to African countries were found, only across 50 African countries with eight not mentioned at all. The most significant country with no mentions was the Ivory Coast (Cote d'Ivoire), followed by Western Sahara. The remainder were small island states. As with the authorship bias towards South Africa, a significant majority of GIGGA research is South Africa focused. Out of all the countries in Africa, South Africa featured in 382 citations (40% of the total). This is equivalent to all the citations featuring the next six most commonly mentioned African countries (Kenya, Tanzania, Ghana, Nigeria, Ethiopia and Uganda) and 20 more than the bottom 50 African countries combined. Twenty-seven African countries featured in less than 1% of the citations. Figure 3 graphically illustrates the patchy distribution of study focus. Note that the difference between South Africa and the next nearest countries is not effectively represented in this figure.

### 4.4. Topical Focus of GIGGA Research

#### 4.4.1. Most Commonly Occurring Terms

Out of all the 55 search terms generated to capture the meanings under each element of GIGG, the term that featured in most citations was 'develop*', found in 808 citations, 84% of the total. This is

likely in part due to the twin uses of this route that can refer both to 'development' as in economic growth as well as all the simpler meanings around emerge or create over time. The next four most commonly occurring terms were all found in a third or more of the 960 citations. Both 'policy' and 'environment\*' were found in more than 40% of the citations (48% and 45% respectively) and 'government' and 'rural' in more than 30% (34% and 33% respectively). This is interesting given the focus for inclusion was more on green growth than on governance or sectors by which that could be achieved.

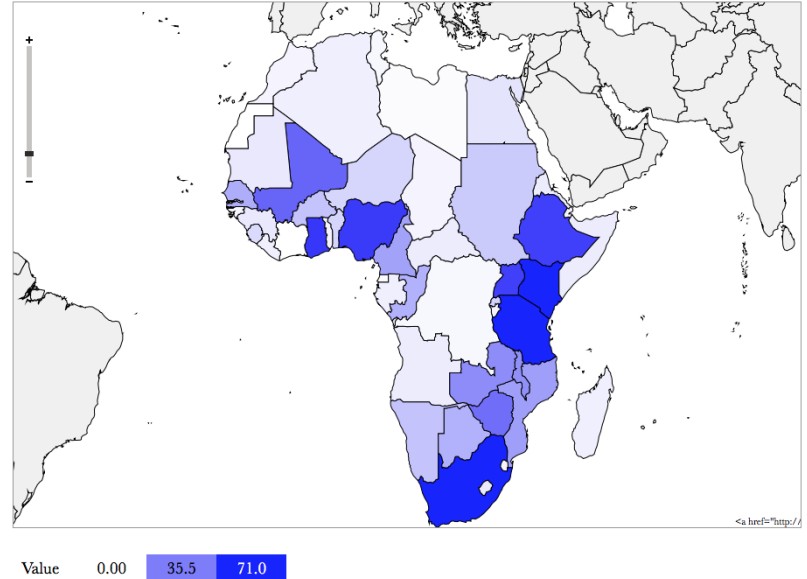

**Figure 3.** Schematic heat map of African country focus of GIGGA citations shown in a standard map of Africa for ease of interpretation. The darker the colour blue the more citations with that country focus.

### 4.4.2. Least Commonly Occurring Terms

Terms that hardly featured at all – here defined as appearing in 1% or less of the citations–are set out in Table 2.

**Table 2.** GIGG search terms featuring in 1% or less of the citations.

| Search Term | Citations Appeared in | % of Total Citations |
|---|---|---|
| Quality of life | 13 | 1% |
| Public sector | 10 | 1% |
| REDD* | 10 | 1% |
| energy security | 9 | 1% |
| Living standards | 9 | 1% |
| Fossil fuel | 8 | 1% |
| GDP | 8 | 1% |
| Social responsibility | 7 | 1% |
| water security | 6 | 1% |
| Grabbing | 5 | 1% |
| Low Income | 5 | 1% |
| Natural capital | 5 | 1% |
| Externalit* | 3 | 0% |
| co-operative | 2 | 0% |
| INDC* | 2 | 0% |
| Local authority | 1 | 0% |
| Social enterprise | 1 | 0% |
| GNP | 0 | 0% |
| green capital | 0 | 0% |



In retrospect, some of these terms are not surprisingly rare: INDCs had only recently been established at the point of doing the research and some terms are perhaps parochial or particular ('local authority', 'grabbing', 'green capital'). However, the lack of mention of basic growth and economic terms such as GDP, externalities, living standards, low income, and the increasingly popular notion of 'natural capital' as well as widely-used resource policy terms such as 'water and energy security' point to a distinct discourse for GIGGA separate from wider global academic discourses. Similarly, terms which are relatively high frequency in the Scopus database such as 'quality of life' (over 14,000 hits using the same exclusion criteria as here) and 'public sector' (over 17,000 hits) reveal a particular gap in the way GIGGA research is framed compared to research more generally.

### 4.4.3. Comparing GIGG Concepts Overall

Originally the 55 search terms served to describe the individual elements of GIGG, so understanding their grouped effect is important to understand whether there are any imbalances or gaps at the GIGG level. Comparing between the groups of terms under the GIGG concepts is made more difficult by virtue, *inter alia*, of there being different number of search terms used for Governing (9) compared with Inclusive (18), Green (17) or Growth (17). This in part is because the obvious way to determine whether one area or another is the subject of greater focus is to look at the total hits for each group. Doing so is likely to bias those with greater numbers of terms. Averaging the number to address this will bias the analysis to those with fewer terms. All this is notwithstanding any effects of frequent use of wildcards which increases the effective potential hit rate.

Across all the 55 GIGG search terms, 5339 hits were recorded—that is, the sum overall for the number of citations each term was found in. To this, Governing contributed the least (1249, 24%) and Green the most (1499, 28%). This distribution is graphically charted in Figure 4. However, as noted above Governing has half the number of terms compared with Green. If we take just the top four terms that recorded the highest number of hits for each GIGG element, thereby mitigating the effect of number of search terms, we find almost the reverse picture. Here Growth comes out on top with 1190 hits, Governing second with 1119 hits and Green third with 1022 hits.

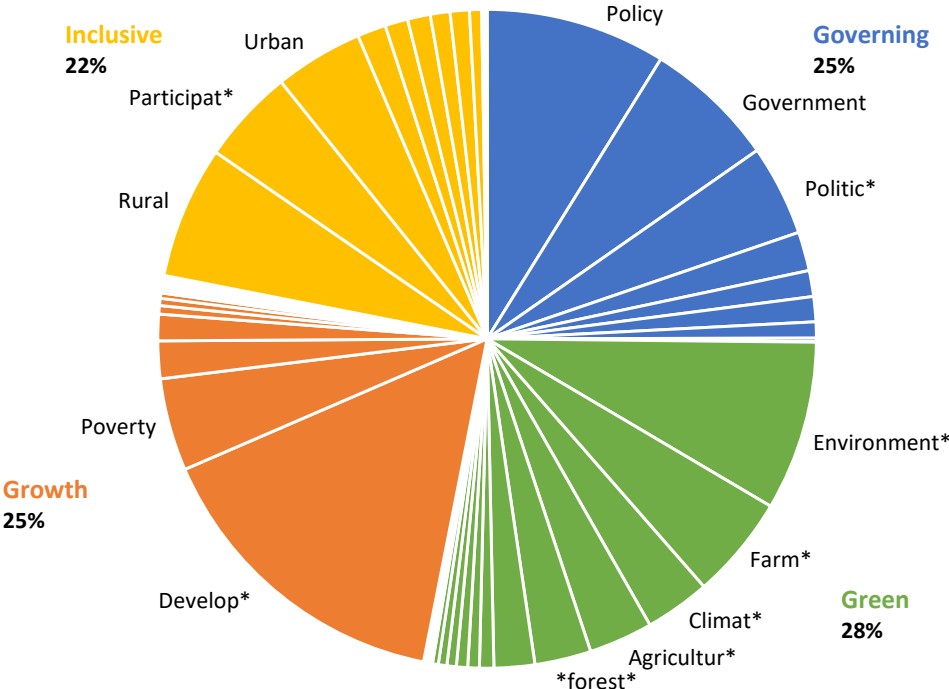

**Figure 4.** Pie chart showing distribution of search term hits within GIGG categories.

What seems to hold regardless of the analysis is that Inclusive is the area with least focus. Within this group, the term the terms 'rural', 'participat*', 'poverty' and 'urban' are the most common. Between then, they appear on average in around a quarter of the GIGGA citations. This represents a gap or weakness of sorts, but perhaps a relative one. More telling perhaps is the lack of occurrence of terms related to mechanisms for improving inclusiveness ('social enterprise', 'social responsibility', 'co-operative', 'equity'), indicating that the real gap in inclusion research for GIGGA is around the mechanisms for inclusion. Similarly, particular groups indicated by a limited occurrence of terms like 'gender' (6% of citations) and 'women' (7% of citations) also suggest a need for greater research efforts, alongside outcome discourses that likely have greater relevance to such groups ('well-being', 'quality of life' and 'living standards'—all found in less than 2% of the GIGGA citations).

## 5. Discussion

It is widely stated that the green economy has significant potential for delivering inclusive and sustainable economic development in Africa. However, research on green growth around the Continent is very limited, with the result that the dynamics of this potentially paradigm-shifting phenomenon are still poorly understood. In this article our starting point was that having a greater quantity and quality of data available on green growth will facilitate identification of research gaps and also improve the ability to make informed decisions regarding the governance of green growth. This led us to draw on Cash et al.'s [21] argument that it is important to mobilize knowledge systems for sustainable development, being cognizant of the interaction between research and governance that feeds into the effective creation of knowledge for policy making. Framing the evidence-gathering and discussion with respect to three characteristics of scientific information production that are needed for research to support and influence policy making—salience, legitimacy, and credibility—we undertook a systematic mapping exercise of the academic literature using the Scopus database.

The orientation of this exercise was towards elucidating the geographical and topical focus on academic publication across the continent, underpinned by the need to understand where the strongest resource bases are for a credible supply of salient research across the Continent. Hence our methodology focused on what countries were producing research on Africa, which countries this research focused on and the topic areas; this was considered more appropriate than statistical meta-analysis on journal publication patterns.

With regard to geographical locus, our underlying hypothesis presupposed that Africa-based authors would be in a better position to undertake policy-relevant research on African green growth issues than those outside Africa, due to their contextual understanding. Our results revealed that 48% of associations were with African countries and 52% were non-African. With South Africa-based authors represented on 32% of all citations in the GIGGA database there was a clear geographical bias for GIGGA expertise on the Continent. Non-African associations were dominated by the UK (12% of citation associations, 16% of all citations) and the US (9% and 13% respectively). Over 60% of all the citations in the database carried authors based in South Africa, UK or the US. Within Africa but beyond the obvious focus of South Africa, Tanzania/East Africa and Nigeria having growing hubs of expertise.

Probing the geographic location of authors' institutions to know how many were all based either within African countries or outside, or alternatively in combination within and without, gave the overall finding that African-based authors were involved in the majority of citations (75%) suggesting a positive basis for building continental knowledge production. However, South Africa skews this picture significantly, if South African authors are omitted from the analysis the results show that the majority of GIGGA research is based on authors outside the Continent (386 citations, 58%). Clearly this raises concerns for the development of robust knowledge production systems within the Continent.

Turning from author geographical base to topical focus of GIGGA research, the study was predicated on the assumption that research is needed on specific countries in order to benefit from the application of expertise. Moreover, from the institutional geography emerges particular perspectives on African countries GIGGA research. Consistently, given the authorship bias already identified,

South Africa dominates GIGGA research, equivalent to Kenya, Tanzania, Ghana, Nigeria, Ethiopia and Uganda combined, while at the other end of the scale Ivory Coast (Cote d'Ivoire), Western Sahara and small island states featured no mentions.

The distribution of citations according to GIGG—governance, inclusion, green, growth—reveals few surprises for knowledgeable observers in this field, with 'develop*' found in 84% of the total (808 citations), followed by 'policy' and 'environment*' (48% and 45% of citations respectively) and 'government' and 'rural' (34% and 33% respectively). Other terms within the development field were notably absent - GDP, externalities, living standards, low income, 'natural capital' and resource policy terms such as 'water and energy security'. This all points to a distinct discourse for GIGGA separate from wider global development discourses.

Terms captured under the key word 'inclusive' proved the area with the least focus, though within this group, the term the terms 'rural', 'participat*', 'poverty' and 'urban' are the most common. Between then, they appear on average in around a quarter of the GIGGA citations. This represents a gap or weakness of sorts, but perhaps a relative one. More telling perhaps is the lack of occurrence of terms related to mechanisms for improving inclusiveness ('social enterprise', 'social responsibility', 'co-operative', 'equity'), indicating that the real gap in inclusion research for GIGGA is around the mechanisms for inclusion. Similarly, particular groups indicated by a limited occurrence of terms like 'gender' (6% of citations) and 'women' (7% of citations) also suggest a need for greater research efforts, alongside outcome discourses that likely have greater relevance to such groups ('well-being', 'quality of life' and 'living standards'—all found in less than 2% of the GIGGA citations).

A key criticism of the study includes the degree to which the search terms predefine the issues looked at. To an extent this is certainly true, and necessarily so. The consequence of this is generate significant false negatives, implying there is less, and less relevant research available in GIGGA topics than actually exists. It is certainly the case that research will have been missed, but the key point is that any terms not used to describe these phenomena are likely terms used outside the mainstream of academic thought both in the global North and in Africa—since a majority of the team are from African countries and based in African countries. Also, the use of Scopus clearly demarcates a particular notion of 'academic research' which certainly excludes journals not indexed by Scopus. At the same time, this demarcation is intentional if problematic: if the global North sets the de facto standards for what 'credible' research is, this is likely to favour historically contingent modes of knowledge production set up by global North institutions favouring the global North. Yet, in the absence of other simple means of establishing quality criteria, this kind of research would not be possible. Instead it perhaps serves as an indication that a programme of defining quality relevant to African research is needed as part of a wide programme of promoting more African-based, African-led research into green growth.

## 6. Conclusions

In 1987 Gro Harlem Brundtland wrote:

*"Scientists bring to our attention urgent but complex problems bearing on our very survival: a warming globe, threats to the Earth's ozone layer, deserts consuming agricultural land. We respond by demanding more details, and by assigning the problems to institutions ill-equipped to cope with them."* [17]

The key question we have addressed here is which "scientists", where are they based, and about what places and topics do they research? And it is clear from the work of Cash et al. [21] that the answer should include a significant number of African, and Africa-based scientists focusing on complex African problems, including how to equip African institutions to cope with them. As our work reveals, while progress has been made in Africa a majority of the Continent risks being left behind. Further, it is not just the absence of research about particular regions, but also gaps on key topics, at least as they are defined by the wider research field. Filling these gaps will be an important step in strengthening policy-making in line with the GIGG concepts.

Overall, and with a policy focus in view, the conclusions that can be drawn from this systematic exercise are, in some sense, unsurprising, namely there needs to be greater investment in research on inclusive green growth within African countries. Giving attention to improving strategic research links across the Continent and between southern and northern institutions is perhaps the most viable way to proceed in an endeavor to increase Africa's research power. The research here provides a possible starting point however in both the foundations of that research capacity sitting across 4 or 5 African nations, and a clear sense of what topics might be considered a priority. In terms of salience of research for policy, ensuring these research links enable policy makers and academics to work together could contribute to building pathways through which knowledge production systems effectively inform decision-making by government and associated actors.

**Author Contributions:** Conceptualization, A.C., E.F., Y.M. and C.O.; Data curation, A.C.; Formal analysis, A.C., C.M., E.F., Y.M., M.G., M.O., A.-B.M., K.C.A. and C.O.; Funding acquisition, C.O.; Methodology, A.C. and C.M.; Validation, A.C., C.M., E.F., Y.M., M.G., M.O., A.-B.M., K.C.A. and C.O.; Writing—original draft, A.C., C.M., E.F., Y.M., M.G., M.O., A.-B.M., K.C.A. and C.O.; Writing—review & editing, A.C., C.M., E.F., Y.M., M.G., M.O., A.-B.M., K.C.A. and C.O. All authors have read and agreed to the published version of the manuscript.

**Funding:** This research was funded by the Economic and Social Research Council of the United Kingdom under the Global Challenges Research Fund, grant number ES/P006671/1. The APC was funded by the University of Reading.

**Acknowledgments:** We would like to express our sincere thanks to all network members and advisory board of the GIGGA project and also to invited guests for invaluable reflection on the issues contained in this paper during the course of workshops held in Kenya, Nigeria, Ethiopia and the UK.

**Conflicts of Interest:** The authors declare no conflict of interest. The funders had no role in the design of the study; in the collection, analyses, or interpretation of data; in the writing of the manuscript, or in the decision to publish the results.

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
