# Peer review of "Mapping Academic Literature on Governing Inclusive Green Growth in Africa: Geographical Biases and Topical Gaps"

_sustainability, doi:10.3390/su12051956_

Round 1
Reviewer 1 Report
Dear Authors, I enjoyed reading this paper, thank you. I would suggest you to improve two specific points of the manuscript.
1) The specific thresholds and pre-requisite of the analysis should be better detailed in paragraph 3.1.1. Inclusion and exclusion of specific issues/terms/words in the qualitative research should be better motivated andf possibly justified with reference to previous scientific works. If not available, an enriched rationale discussing pros & cons of your criteria should be proposed.
2) generally speaking, a better connection with the traditional literature of developmental economics should be provided in the introduction and commented a bit in the discussion section. What is the take-home message of the study? This point should be better elaborated in a revised version.
Reviewer 2 Report
The paper deals with the mapping of literature on governing inclusive green growth in Africa (GIGGA). The topic is rather interesting and it fit with the aim of the journal. Unfortunately the manuscript seems as a report rather a scientific paper. The introduction and presentation of the research can be acceptable but the methodology is not absolutely clear. I don't understand the novelty of the paper there a simple presentation of data acquired from Scopus and Web of Science. For this reason I suggest to: 1)define clearly the aim of the paper, 2)rewrite the section concerning the methodology and the results, 3) re-submit as review paper.
Author Response
AUTHORS' REMARKS ON THE REVIEWER:
We want to thank the reviewer for taking time to look at the paper and offering comments intended to improve the paper and prepare it for publication. We respectfully disagree with the judgement implied and have taken care to explore and explain and demonstrate at length why we disagree and how our disagreement is justified. Science advances through disagreement, but only if each sides can present reasoned, evidenced positions. Our position rests on three points:
This is not a review paper in the standard sense. If it were we would agree with the reviewer.
The approach used is recognised as being deployed in other published papers and falls into the 'systematic review/bibliometric/scientometric' category as described in the paper and reflected in the papers reviewed in the introductory sections.
Two other reviewers have agreed in general with the appraoch and
We generally like to take into account what reviewers say, but sometimes that is not possible, especially if we believe the perspective taken on the paper does not match the expressed perspective set out in the paper.
REVIEWER SAYS:
The paper deals with the mapping of literature on governing inclusive green growth in Africa (GIGGA). The topic is rather interesting and it fit with the aim of the journal. Unfortunately the manuscript seems as a report rather a scientific paper. The introduction and presentation of the research can be acceptable but the methodology is not absolutely clear. I don't understand the novelty of the paper there a simple presentation of data acquired from Scopus and Web of Science.
For this reason I suggest to:
1)define clearly the aim of the paper
AUTHORS RESPONSE: We believe this has been done, in response to previous reviewers comments. It is, as it says both in the Abstract and the introductory section, to map the knowledge capacity for green growth in Africa against the criteria for effective knowledge production in order to identify geographic, institutional and topical gaps that need filling to help promote GIGGA. We state in p2, lines 85-6 "We argue here that for Africa to benefit from green growth, not only must that green growth be inclusive it must be underpinned by an effective knowledge system." This of course provides the guiding rationale for wanting to explore how effective the knowledge system in Africa for informing policy on GIGG is, where the strengths and gaps are using the criteria for effectiveness established by Cash et al 2003. This reveals novel insights in what helps, and what's missing where all set out in various parts of the paper.
2)rewrite the section concerning the methodology and the results
We believe this is about as clear as it is possible to be within the confines of an already long paper - neither of the two other reviewers required such a rewrite and it is not clear to us what kind of rewrite would be sufficient to address the concerns expressed. It may be that their are misinterpreting the nature of the paper, as implied by their third requirement below. This paper uses scientometric/bibliographic and systematic review methods to map knowledge production through the proxy data point of academic article production. Further, the description of the method is in line with (if not actually more transparent, systematic, well-reasoned and clear in part because we used simpler statistical approaches due to the small scale of the data) the other papers cited in the introduction which deploy a similar approach (Kiparksy et al 2006; Confraria & Godinho, 2015; Wang et al. 2017; Pasgaard et al 2015).
3) re-submit as review paper.
This is not a 'review' paper in the standard sense that the reviewer seems to imply, where the author or authors narratively reviews the literature in a topical area - it is a bibliometric study of knowledge production using standard methods for this field. This differs from the normal 'narrative review' in being replicable and transparent and justified in terms of search strategy, search tools, inclusion and exclusion criteria and quantitative in the mapping of content. Hence the reason we review other papers using a similar method in this area such as those cited above. As such this qualifies as a research paper rather than a review paper on account of it taking an empirical approach to a dataset (here, published academic papers in the Scopus database) similar to e.g. analysing any other quantitative dataset (such as weather patterns, car movements, survey responses etc).
To return to the reviewers question of why this is novel, the answer is for the same reason any other bibliometric study of the patterns of academic research is novel, indeed for the same reasons any research study of the patterns of any data is novel: because in doing so a) no-one has examined that dataset in that way before; b) it is interesting because it builds on established concepts, frameworks or theories; c) because it reveals potentially useful orientations for effective action.
Our paper fulfills a-c, as much as any other paper cited. It is 'simple' in the sense that we do not use complex mathematical/statistical approaches such as deployed by Confraria and Godinho, but that is a consequence of the smaller numbers that mean characterisations of the patterns can be done directly from the counts without resorting to complicated transformations.
As a consequence, we feel the reviewer has misunderstood the nature of this kind of inquiry and in doing so has concluded that it lacks merit, or it's merit can only be established via repositioning the paper in an approach they recognise. We respectfully reject that analysis and provide reasons for doing so.
best wishes
The authors
Reviewer 3 Report
The manuscript “Building a Framework of Evaluating Human-Environmental Relationships: Considering the Differences of Subjective Evaluation and Objective Assessment” presents a very important issue of academic achievement. They are an important indicator of the ability to create knowledge. Research based on the GIGGA example.
The manuscript is a methodical study.
Comments for authors
Language
There are no remarks to the language.
Subject
It corresponds to the content.
Keywords
The keywords have been chosen correctly
Abstract
The summary partly reflects the content of the manuscript. It does not contain the purpose of the work, which is a major mistake. The aim of the study was determined indirectly. The abstract should be edited.
Introduction
I have no general remarks to this part. The literature review is correct but not so rich. Research questions defined correctly and logically.
In the introduction, the authors refer only to Africa. Sustainability Journal is an international journal and a broader research context should be provided.
The last paragraph should be removed. It is not necessary to provide the manuscript structure. The preferred style is described in the guide for authors.
Knowledge systems for governance of inclusive green growth in Africa
I have no objections to this part.
Methods
Description of the methodology is very general. An oceral scheme could be useful to a wider audience. I recommend thinking about the possibility of its development.
3.1. Search strategy
A fully informative subsection, consistent with the purpose of the research. According to the reviewer, the choice of analyzed bases is correct.
3.1.1 Scope and Exclusions
Correct section.
3.1.2 GIGA term identification
Correct description of the research..
3.1.3 World search terms
The strategy for the wider search is the logical research consequence. I have no objections to this part of the manuscript.
3.1.4 Search term
3.1.5 Data handling and verification
I have no major objections to these parts. Due to the large fragmentation of the subsections in this part, I suggest combining them.
3.1.6 Mapping the content and meta data
Correct section.
Results
Division of subsections too detailed (4.1.1 and 4.1.2). The description is correct, but for the second time there is a suggestion to develop the research scheme.
This section contains many discussion elements. I suggest two solutions:
Title the section – Results and discussion. In this case, move some content from Chapter 5. Or separate the results and discussion. Also move the discussion part from chapter 5.
This arrangement will provide better readability.
The figures 1 and 2 are the maps. They are incorrect in cartographic terms. The north arrow and the linear scale are missing, and the legend is outside the map drawing.
Discussion and conclusions
The discussion is correct related to similar studies. It must be moved absolutely to chapter 4 or a separate chapter should be introduced.
The conclusion part must be a separate chapter.
Final remarks
I think that the presented manuscript is a valuable study that concerns a very difficult issue. The comments are disputable. I hope that including them in the final version will increase its scientific value.
Conclusion from the review – the manuscript requires minor changes recommended by the reviewer.
Round 2
Reviewer 1 Report
Good revision, I agree with the new version, it is ready to be published.
Author Response
We would like to thank the reviewer for taking the time to provide useful and insightful input.